# Optimized Unconventional Geometric Gates in Superconducting Circuits

Yueheng Liu  and Xinding Zhang *

Guangdong Provincial Key Laboratory of Quantum Engineering and Quantum Materials,
School of Physics and Telecommunication Engineering, South China Normal University,
Guangzhou 510006, China
* Correspondence: xdzhang@scnu.edu.cn; Tel.: +86-1363-244-0734

**Abstract:** Nonadiabatic Abelian geometric quantum computation has been extensively studied, due to its fast manipulation and inherent noise resistance. However, to obtain the pure geometric phase, the quantum state is required to evolve along some special paths to eliminate the dynamical phase. This leads to increasing evolution time and weakened gate robustness. The unconventional geometric quantum computation is an effective way to solve the above problems. Here, we propose a general approach to realize the unconventional geometric computation. Then, we discuss the effect of the ratio of geometric phase to dynamic phase on the performance of quantum gates. The results show that the selection of ratio corresponds to different quantum gate robustness. Therefore, we can optimize the ratio to get higher-fidelity quantum gates. At last, we construct the ratio-optimized quantum gates in a superconducting circuit and test its robustness. The fidelities of the T-gate, Hadamard H-gate, and controlled phase gate can be obtained as 99.98%, 99.95%, and 99.85%, respectively. Therefore, our scheme provides a promising way to realize large-scale fault-tolerant quantum computation in superconducting circuits.

**Keywords:** geometric quantum computation; unconventional geometric quantum computation; high-fidelity quantum gates; superconducting circuits



## 1. Introduction

Quantum computers can solve some complex problems that the classical ones cannot solve [1,2], because of the coherence and entanglement of the quantum process. Constructing quantum computers requires a set of universal quantum gates, including a set of arbitrary one-qubit gates and a nontrivial two-qubit gate [3,4]. However, the decoherence from the surrounding environment and the noise effects decrease the quantum gate fidelity. Thus, it is essential and challenging to construct quantum gates that have a high enough gate fidelity and a strong robustness against errors.

In 1984, Berry found the geometric phase in an adiabatic process, which was an element of the holonomy group defined by the connection in the parameters space [5]. Meanwhile, the geometric phase only depends on the geometric properties of the curve in the parameter space [6]. Hence, the quantum gates constructed by the geometric phase have a strong robustness against certain local errors [7–9]. However, the adiabatic condition requires the quantum system be exposed to the environment for a long time. In 1987, Aharonov and Anandan generalized the Berry phase by relaxing the adiabatic condition [10]. The nonadiabatic geometric quantum computation (NGQC) needed less evolution time due to the nonadiabatic process. Since then, NGQC has been extensively studied theoretically [11–14] and experimentally implemented in various quantum systems [15–17]. Furthermore, the Abelian NGQC can be promoted to the non-Abelian case [18], and has also been widely used to generate a quantum computation [19–23].

The important task of geometric quantum computing is how to deal with dynamic phases that are not protected by geometric properties. There are two common approaches.

The first is to design a special evolutionary path to completely eliminate the dynamic phase [24–26], which often requires a long evolutionary distance, thus exacerbating the decoherence effect caused by the external environment. For example, no matter how small the rotating gate is, the evolution path required in the orange-slice-shape-path (OSSP) scheme is formed by two lines of longitude.The second approach is unconventional geometric quantum computing, which manages to convert dynamical phases into geometric features. However, this method requires a complex system parameter design. Moreover, the ratio of the kinetic phase to the geometric phase is difficult to adjust in the latter scheme [27,28].

Now, we ask two questions. Inspired by general methods for producing nonadiabatic and holonomy quantum computations, whether we can also find a general approach to realize an unconventional geometric quantum computation [29,30]. Here, the ratio of the dynamical phase to the geometric phase is arbitrarily adjusted. Assuming such a method exists, would different ratios affect the robustness of the quantum gates?

In this work, we derive a general approach to implement quantum computing with unconventional geometries. This method can be understood as a mapping. As long as we input a set of pendant states and path parameters, we can get an accurate Hamiltonian and the ratio can be adjusted arbitrarily. Next, we optimized the ratio to improve the robustness of the quantum gates. Finally, we construct our ratio-optimized single-qubit T-gate, H-gate, and two-qubit controlled phase gate in superconducting circuits. We perform numerical simulations of quantum gates. The results show that the fidelities of the T-gate, H-gate and controlled phase gate reach 99.98%, 99.95%, and 99.85%, respectively. Therefore, our scheme provides a promising way to realize large-scale fault-tolerant quantum computation in superconducting circuits.

## 2. General Approach

### 2.1. Geometric Phase

Two states are said to be physically equivalent ($|\psi\rangle \sim |\phi\rangle$) if $|\psi\rangle = e^{i\alpha}|\phi\rangle$ with a real function $\alpha$. Therefore, a proper space to describe the quantum dynamic evolution is the projective Hilbert space $\mathcal{P}(\mathcal{H})$ [31]. Here, all states which are equivalent to $|\psi\rangle$ represent an equivalence class in $\mathcal{P}(\mathcal{H})$. Consider an L-dimensional quantum system defined by Hamiltonian $H(t)$. $\{|\psi_i(t)\rangle\}$ represent L orthonormal solutions of the Schrödinger equation $id_t|\psi_i(t)\rangle = H(t)|\psi_i(t)\rangle$. If the trajectory of $|\psi_i(t)\rangle$ is projected to a closed curve $C(t)$ in $\mathcal{P}(\mathcal{H})$, this evolution is called a cycle. After a cyclic evolution, the state $|\psi_i(T)\rangle$ and $|\psi_i(0)\rangle$ project to the same point in the projective space. Hence, we have $|\psi_i(T)\rangle = e^{i\phi}|\psi_i(0)\rangle$ [10].

To calculate the phase $\phi$, we consider a set of auxiliary orthonormal states $|\tilde{\psi}_i(t)\rangle$, $i = 1, ..., L$, which satisfy two conditions, (1) the trajectory of the auxiliary state projects to the same closed curve $C(t)$ in $\mathcal{P}(\mathcal{H})$, i.e., $|\psi_i(t)\rangle$ is equivalent to $|\tilde{\psi}_i(t)\rangle$; (2) the auxiliary state satisfies $|\tilde{\psi}_i(T)\rangle = |\tilde{\psi}_i(0)\rangle$. Then, we can obtain

$$|\psi_i(t)\rangle = e^{if_i(t)}|\tilde{\psi}_i(t)\rangle,$$
$$f_i(T) - f_i(0) = \phi. \tag{1}$$

Substituting Equation (1) into the Schrödinger equation, we obtain

$$f_- = i\int_0^t \langle\tilde{\psi}_i(t')|d_t|\tilde{\psi}_i(t')\rangle \, dt' - \int_0^t \langle\psi_i(t')|H(t)|\psi_i(t')\rangle \, dt', \tag{2}$$

where $f_- = f_i(t) - f_i(0)$. The first term above represents the geometric phase $\gamma_{gi}(|\tilde{\psi}(t)\rangle)$, and the other is the dynamic phase $\gamma_{di}(|\tilde{\psi}(t)\rangle)$.

The choice of auxiliary bases $\{|\tilde{\psi}_i(t)\rangle\}$ is not unique. We consider other states $\{|\psi'_i(t)\rangle\}$ which satisfy the above conditions (1) and (2). We can easily get $|\psi'_i(t)\rangle = e^{i\xi(t)}|\tilde{\psi}_i(t)\rangle = e^{i[\xi(t)-f_i(t)]}|\psi_i(t)\rangle$, and $\xi(T) = \xi(0)$. Substituting $|\psi_i(t)\rangle$ in the Schrödinger equation again, we can obtain the geometric phase $\gamma_{gi}(|\psi'_i(T)\rangle) = \gamma_{gi}(|\tilde{\psi}_i(T)\rangle) - [\xi(T) - \tilde{\xi}(0)] = \gamma_{gi}(|\tilde{\psi}_i(T)\rangle)$. Therefore, the geometric phase $\gamma_{gi}(|\tilde{\psi}(t)\rangle)$ is $U(1)$-gauge-invariant. Note

that the above derivation can be applied to arbitrary $|\psi_i(t)\rangle$ under cyclic evolution and any $|\tilde{\psi}_i(t)\rangle$ which satisfies conditions (1) and (2).

*2.2. Constructing the General Approach*

We start from the state $|\psi_i(t)\rangle = e^{if_i(t)} |\tilde{\psi}_i(t)\rangle$ in Equation (1) with $f_i(0) = 0$. Hence, $|\psi_i(t)\rangle = e^{if_i(t)} |\tilde{\psi}_i(t)\rangle$ is obtained. According to the unconventional geometric scheme, the total phase can be written as

$$f_i(t) = \gamma_{gi}(t) + \gamma_{di}(t) = (1 + \eta)\gamma_{gi}(t), \tag{3}$$

when $\gamma_{di}(t) = \eta \gamma_{gi}(t)$ is satisfied. In this situation, $|\psi_i(t)\rangle = e^{(1+\eta)\gamma_{gi}(t)} |\tilde{\psi}_i(t)\rangle$.

We now ask how to find a Hamiltonian which can make the quantum system evolve from the initial $|\psi_i(0)\rangle$ to the final state $|\psi_i(t)\rangle$, such that Equations (1) and (3) are fulfilled. To find such Hamiltonian, we can substitute Equation (1)–(3) into $H(t) = i\sum_{i=1}^{L} |\dot{\psi}_i(t)\rangle \langle \psi_i(t)|$. After the calculation given in appendix A, we can get

$$H(t) = i\sum_{i \neq j}^{L} \langle \tilde{\psi}_i(t)|\dot{\tilde{\psi}}_j(t)\rangle |\tilde{\psi}_i(t)\rangle \langle \tilde{\psi}_j(t)| - i\sum_{i=1}^{L} \eta \langle \tilde{\psi}_i(t)|\dot{\tilde{\psi}}_i(t)\rangle |\tilde{\psi}_i(t)\rangle \langle \tilde{\psi}_i(t)|. \tag{4}$$

The above derivation shows that starting from an arbitrary basis $\{|\tilde{\psi}_i(t)\rangle\}$, $i = 1, ..., L$, with $|\tilde{\psi}_i(0)\rangle = |\tilde{\psi}_i(T)\rangle$, we can obtain the accurate Hamiltonian by using Equation (4) such that the state $|\psi_i(t)\rangle$ defined by Equations (1) and (3) is the solution of the Schrödinger equation. Consequently, the evolution operator at time T can be written as

$$U(T) = \sum_{i=1}^{L} e^{if_i(T)} |\tilde{\psi}_i(0)\rangle \langle \tilde{\psi}_i(0)|; \tag{5}$$

in summary, the general Hamiltonian Equation (4) is like a machine. When we input an arbitrary set of auxiliary states $|\tilde{\psi}_i(t)\rangle$, this machine will output a Hamiltonian such that $U(t) |\tilde{\psi}_i(0)\rangle = e^{if_i(t)} |\tilde{\psi}_i(t)\rangle$ is filled.

## 3. Quantum Gates and Robustness

To realize an arbitrary unconventional geometric quantum gate, we consider a two-level system consisting of qubit states $\{|0\rangle, |1\rangle\}$. One choice of the auxiliary bases can be

$$|\tilde{\psi}_1(t)\rangle = \cos\frac{\alpha(t)}{2} |0\rangle + \sin\frac{\alpha(t)}{2} e^{i\beta(t)} |1\rangle,$$
$$|\tilde{\psi}_2(t)\rangle = -\sin\frac{\alpha(t)}{2} e^{-i\beta(t)} |0\rangle + \cos\frac{\alpha(t)}{2} |1\rangle, \tag{6}$$

and these states are required to fulfill the cyclic condition $|\tilde{\psi}_i(0)\rangle = |\tilde{\psi}_i(T)\rangle$. Accordingly, the geometric phase is obtained as

$$\gamma_{g2}(t') = -\gamma_{g1}(t') = i\int_0^t \langle \tilde{\psi}_2(t')|d_t|\tilde{\psi}_2(t')\rangle \, dt' = \frac{1}{2}\int_0^{t'} [1 - \cos\alpha(t)]\dot{\beta}(t) \, dt; \tag{7}$$

when the cyclic evolution condition is satisfied, the above integral can be converted into an area integral using Green's formula. At this point the geometric phase is dependent on the solid angle enclosed by the closed path. For brevity, we redefine $\gamma_{g2}(t) = \gamma_g(t)$, and $f_2(t) = -f_1(t) = f(t) = (1 + \eta)\gamma_g(t)$. The evolution operator for the auxiliary states above can be written as

$$U(T) = e^{-if(T)} |\tilde{\psi}_1(0)\rangle \langle \tilde{\psi}_1(0)| + e^{if(T)} |\tilde{\psi}_2(0)\rangle \langle \tilde{\psi}_2(0)| = e^{-if(T)\boldsymbol{n}\cdot\boldsymbol{\sigma}}, \tag{8}$$

where $\boldsymbol{n} = (\sin\alpha_0 \cos\beta_0, \sin\alpha_0 \sin\beta_0, \cos\alpha_0)$ with $\alpha_0 = \alpha(0)$, $\beta_0 = \beta(0)$, and $\boldsymbol{\sigma}$ is the Pauli vector. The evolution operator $U(T)$ represents an arbitrary one-qubit gate with an arbitrary rotating axis $\boldsymbol{n}$ and an arbitrary rotating angle $2f(T)$.

Now, we start to construct the one-qubit T-gate and H-gate and discuss the effect of the ratio on the robustness of the quantum gates. The parameters for constructing quantum gates can be set as

$$T - gate : (\alpha_0, \beta_0) = (0, 0), \ f(T) = \pi/8,$$
$$H - gate : (\alpha_0, \beta_0) = (\pi/4, 0), \ f(T) = \pi/2. \tag{9}$$

The exponential maps of the T-gate and H-gate are $U(T) = \exp(-i\pi\sigma_z/8)$ and $U(T) = \exp(-i\pi(\sigma_z + \sigma_x)/2\sqrt{2})$, respectively. In order to make the phase $f(T)$ positive, we choose a counterclockwise evolution direction of the quantum state, i.e., the derivative of the azimuth angle is $\dot{\beta}(t) > 0$.

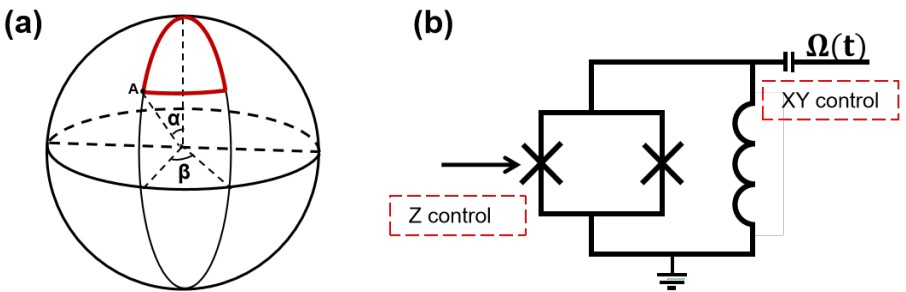

**Figure 1.** (**a**) Schematic diagram of quantum state evolution; the curve is composed of two longitude lines and a section of latitude line, where $\alpha$ and $\beta$ are the polar and azimuth angles, respectively. (**b**) A single transmon is driven by a microwave; we can adjust the Rabi frequency through the xy control line and adjust the detuning through the z control line.

The evolution path of the quantum state is shown in Figure 1a. First, we can take the path parameters $(\alpha(t), \beta(t))$ starting from the north pole $(0, 0)$ to the point A $(\pi/4, 0)$ along the geodesic $\beta(t) = 0$ for $[0, T_1]$. For this path, the Hamiltonian can be obtained, $H(t) = \frac{1}{2}\Omega(t)[e^{i(\beta(t)+\pi/2)}|1\rangle\langle 0| + \text{H.c}]$, with the Rabi frequency being $\Omega(t) = \dot{\alpha}(t) > 0$. The Rabi frequency area is calculated as $\int_0^{T_1} \Omega(t)\,dt = \pi/4$.

Then, the quantum state evolves from point A $(\pi/4, 0)$ to $(\pi/4, \pi/[2(1+\eta)(2-\sqrt{2})])$ along the latitude line $\alpha(t) = \pi/4$, for $(T_1, T_2]$, with $\eta < \frac{\sqrt{2}}{2-\sqrt{2}} (\approx 2.41)$. For this path, the Hamiltonian reads $H(t) = \frac{1}{2}\Delta(t)(|1\rangle\langle 1| - |0\rangle\langle 0|) + \frac{1}{2}\Omega(t)[e^{i(\beta(t)+\pi)}|1\rangle\langle 0| + \text{H.c}]$, where the Rabi frequency and the detuning are written as

$$\Omega(t) = \frac{\sqrt{2}}{2}[\frac{\sqrt{2}}{2}(1+\eta) - \eta]\dot{\beta}(t),$$
$$\Delta(t) = -2[1 + \frac{\sqrt{2}}{2}(1+\eta)]\sin^2(\frac{\pi}{8})\dot{\beta}(t). \tag{10}$$

Note that we took $\eta < \frac{\sqrt{2}}{2-\sqrt{2}}$ above just to make the Rabi frequency $\Omega(t)$ positive. Accordingly, the Rabi frequency area is $\int_{T_1}^{T_2} \Omega(t)\,dt = \frac{\sqrt{2}}{2}[-\eta + \frac{\sqrt{2}}{2}(1+\eta)](\beta(T_2) - \beta(T_1))$. Meanwhile, the ratio must be $-1 < \eta$ to ensure that $\dot{\beta}(t) > 0$. If $\eta > 2.41$, the Hamiltonian reads $H(t) = \frac{1}{2}\Delta(t)(|1\rangle\langle 1| - |0\rangle\langle 0|) + \frac{1}{2}\Omega(t)[e^{i\beta(t)}|1\rangle\langle 0| + \text{H.c}]$ and the Rabi frequency is $\Omega(t) = \frac{\sqrt{2}}{2}[\eta - \frac{\sqrt{2}}{2}(1+\eta)]\dot{\beta}(t)$. If and only if $\eta = 2.41$, we have $\Omega(t) = 0$.

Finally, the state returns back to the north pole $(0, \pi/[2(1+\eta)(2-\sqrt{2})])$ along the geodesic $\beta(t) = \pi/[2(1+\eta)(2-\sqrt{2})]$ for $(T_2, T_3]$. Accordingly, the Hamiltonian is expressed as $H(t) = \frac{1}{2}\Omega(t)[e^{i(\beta(t)-\frac{\pi}{2})}|1\rangle\langle 0| + \text{H.c}]$ with $\Omega(t) = -\dot{\alpha}(t) > 0$. The Rabi fre-

quency area is also $\pi/4$ and the solid angle is calculated as $2\gamma_g(T) = \frac{\pi}{4(1+\eta)}$. Finally, the total phase is $f(T) = (1+\eta)\gamma_g(T) = \frac{\pi}{8}$.

The Hamiltonian along every piecewise path can be described as a two-level system and is driven by the Rabi frequency and the detuning,

$$H(t) = \frac{1}{2} \begin{pmatrix} -\Delta(t) & \Omega(t)e^{-i\mu(t)} \\ \Omega(t)e^{i\mu(t)} & \Delta(t) \end{pmatrix}. \tag{11}$$

The process of constructing the H-gate is very similar to that of the T-gate. For simplicity, the evolution parameters and the Rabi frequency are given directly below:

(1) $(\frac{\pi}{4}, 0) \rightarrow (0, 0)$,

(2) $(0, \frac{1}{1+\eta}\frac{2\pi}{\sqrt{2}-2}) \rightarrow (\frac{\pi}{4}, \frac{1}{1+\eta}\frac{2\pi}{\sqrt{2}-2})$,

(3) $(\frac{\pi}{4}, \frac{1}{1+\eta}\frac{2\pi}{\sqrt{2}-2}) \rightarrow (\frac{\pi}{4}, 0)$.

The Rabi frequency reads $\Omega(t) = -\dot{\alpha}(t)$ in $[0, T_1]$, and $\Omega(t) = \dot{\alpha}(t)$ in $(T_1, T_2)$. Meanwhile, $\Omega(t) = \frac{\sqrt{2}}{2}[\frac{\sqrt{2}}{2}(1+\eta) - \eta]\dot{\beta}(t)$ when $\eta < 2.41$, and $\Omega(t) = \frac{\sqrt{2}}{2}[\eta - \frac{\sqrt{2}}{2}(1+\eta)]\dot{\beta}(t)$ when $\eta > 2.41$ in $(T_2, T_3)$.

We next take $\Omega(t) = \Omega_M \sin(\frac{\pi t}{T_{ij}})$ with $\Omega_M = 20 \times 2\pi$ MHz and $T_{ij} = T_j - T_i$ being the total time of every piecewise path, to evaluate the gate robustness of the implemented T-gate and H-gate in our scheme under decoherence, using the Lindblad master equation [32]

$$\dot{\rho} = -i[H(t), \rho] + \frac{1}{2}\kappa_z \mathcal{L}(\mathcal{A}_z) + \frac{1}{2}\kappa_- \mathcal{L}(\mathcal{A}_-), \tag{12}$$

where $\rho$ is the density matrix of the input state, $\mathcal{L}(\mathcal{A}) = 2\mathcal{A}\rho\mathcal{A}^\dagger - \mathcal{A}^\dagger\mathcal{A}\rho - \rho\mathcal{A}^\dagger\mathcal{A}$ is the Lindblad operator for decay operator $\mathcal{A}_- = \sum_{k=1}^{+\infty}\sqrt{k}|k-1\rangle\langle k|$ and dephasing operator $\mathcal{A}_z = \sum_{k=1}^{+\infty}k|k\rangle\langle k|$ [33]. To test the robustness of the T- and H-gate, we set the general input initial state as $|\varphi(t)\rangle = \cos\theta|0\rangle + \sin\theta|1\rangle$. After the action of the Lie group $U(T)$, the final ideal state $|\varphi(T)\rangle$ was $\cos\theta|0\rangle + \exp(i\pi/4)|1\rangle$ and $[(\cos\theta + \sin\theta)|0\rangle + (\cos\theta - \sin\theta)|1\rangle]/\sqrt{2}$, respectively. The gate fidelity was defined as $F_G = \frac{1}{2\pi}\int_0^{2\pi}\langle\varphi(T)|\rho|\varphi(T)\rangle\,d\theta$ [34], and the integration was numerically done for the 1001 input states with $\theta$ being uniformly distributed within $[0, 2\pi]$. In Figure 2, we obtained the fidelities of the T-gate and H-gate as a function of the parameters $\eta \in (-1, 2.41) \cup (2.41, 10]$ and the decoherence rate $\kappa_- = \kappa_z = \kappa \in [0, 8] \times 2\pi$kHz. In the areas I, II, III, the fidelities of the quantum gates reached $F \leq 99.95\%$, $99.95\% < F < 99.98\%$, and $F > 99.98\%$, respectively. We can see that the white curve is an increasing function as the ratio increases in $(-1, 2.41)$ and is a decreasing function as the ratio decreases $(2.41, 10)$. This means that the fidelity of the quantum gate increased as the ratio approached 2.41 and decreased away from 2.41. Because the Rabi frequency was a sinusoidal pulse, the ratio could not be taken as 2.41. In this article, we mainly discuss the influence of the ratio on the quantum gate. In fact, the polar angle does not necessarily have to be $\pi/4$. For example, we took the polar angle as $\pi/3$ and $\pi/6$. The fidelities of the quantum gates had the same trend as a function of the ratio, and their transition point was 2 and 6.5, respectively.

Here, we give the reason to explain this trend in fidelities versus ratio. As discussed above, when the quantum state evolves along the meridian, the Rabi frequency area is $\pi/4$. The ratio of phases has an effect on the evolution of quantum states along latitude lines. Thus, we plotted the Rabi frequency area along the latitude line as a function of $\eta$ in Figure 3. Clearly, the area of the T-gate (red line) and H-gate (blue line) was reduced when $\eta$ approached 2.41 from the left or from the right. A smaller Rabi area with the same positive Rabi frequency $\Omega(t)$ meant less time for the quantum system to interact with the external environment. Therefore, the decoherence noise was suppressed.

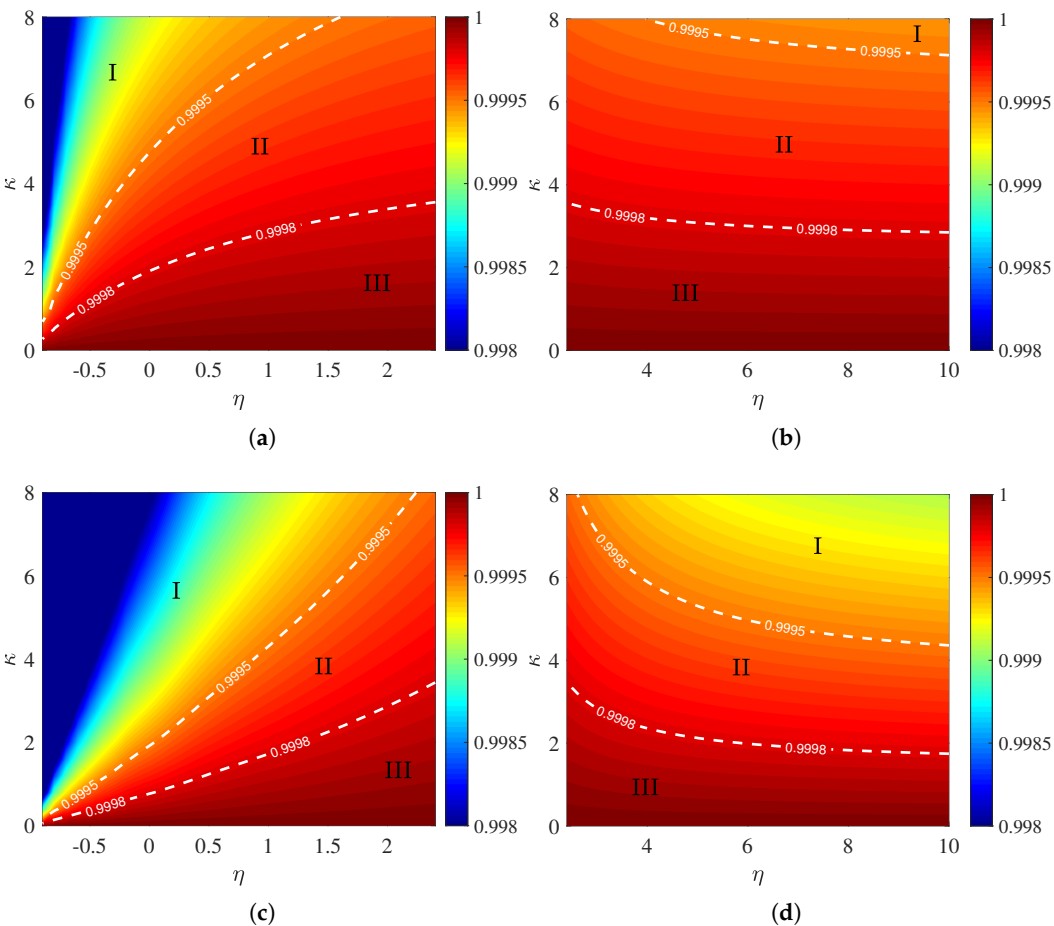

**Figure 2.** Gate fidelities of the T-gate and H-gate as a function of the parameters $\eta$ and $\kappa$, (**a**) $\eta \in (-1, 2.41)$ and (**b**) $\eta \in (2.41, 10]$ for the T-gate, (**c**) $\eta \in (-1, 2.41)$ and (**d**) $\eta \in (2.41, 10]$ for the H-gate. In the areas I, II, and III, the fidelities of the quantum gates reached $F \leq 99.95\%$, $99.95\% < F < 99.98\%$, and $F > 99.98\%$, respectively.

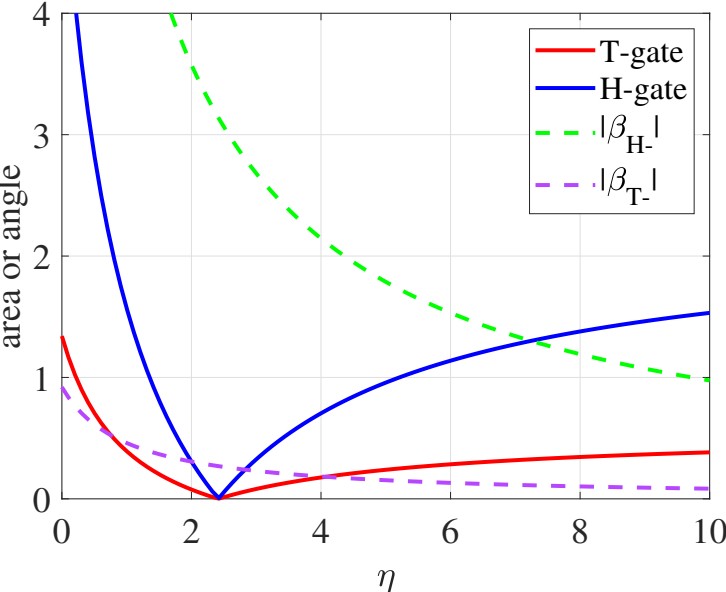

**Figure 3.** The Rabi frequency area along the latitude line, and the change in azimuth $\beta_-$ as functions of the ratio $\eta \in (-1, 10]$.

In the following, we take the ratios of $\eta = 0, 1, 2, 3$ as examples to test their robustness against the control errors, i.e., the qubit-frequency drift $\delta$ and the deviation $\epsilon$ of the driving amplitude, and compare their performance with the OSSP scheme and pure dynamical case. The detailed steps of constructing the quantum OSSP and dynamical gate are in Appendices B and C. The Hamiltonian affected by these controlled errors is written as [33,35]

$$H(t) = \frac{1}{2}\begin{pmatrix} -(\Delta(t) + \delta\Omega_M) & (1+\epsilon)\Omega(t)e^{-i\mu(t)} \\ (1+\epsilon)\Omega(t)e^{i\mu(t)} & \Delta(t) + \delta\Omega_M \end{pmatrix}. \tag{13}$$

The noise $\delta$ appears in the diagonal elements of the matrix, affecting the phase change of the qubit, and $\epsilon$ appears in the off-diagonal elements of the matrix, affecting the flipping between qubits. According to experimental papers [36], we set the decoherence and dephasing ratio as $\kappa_- = \kappa_z = \kappa = 4 \times 2\pi$ kHz. Moreover, we calculated the Rabi area of different ratios. When the ratios were zero, one, two, and three, the areas of the T-gate were $1.34 + \frac{\pi}{2}$, $0.39 + \frac{\pi}{2}$, $0.077 + \frac{\pi}{2}$, and $0.0813 + \frac{\pi}{2}$, respectively, and the areas of the H-gate were $5.36 + \frac{\pi}{2}$, $\pi$, $0.3067 + \frac{\pi}{2}$, and $0.3253 + \frac{\pi}{2}$, respectively. For the dynamical gates, they were $5\pi/4$ for the T-gate and $3\pi/2$ for the H-gate. Moreover, for the OSSP scheme, the areas were all $2\pi$ for the T- and H-gate. It was found that by adjusting the ratio reasonably, the Rabi area could be made smaller than for the OSSP and the dynamic scheme. The robustness comparison of all the above schemes is shown in Figure 4. The results show that when the ratio was close to the phase transition point of 2.41, the robustness of the quantum gate against control errors was indeed improved compared with the pure geometry $\eta = 0$ scheme, the dynamic gate scheme, and the OSSP scheme. The variation of the robustness with the ratio was about the same as previously discussed. In Figure 4d, the fidelity of the $\eta = 1$ H-gate was lower than that of the $\eta = 2$ and $\eta = 3$ gates in $\epsilon \in (-0.04, 0.04)$. It is understandable that the $\eta = 2$ and $\eta = 3$ effects were not much different, because their Rabi areas were not very different.

When constructing quantum gates, we hope that the Rabi area is small enough to increase the robustness of the system against decoherence. Meanwhile, we hope that the change of the azimuth angle cannot be too small, because a small control error in the experiment would lead to a large error in this solution. The gain effect of $\eta = 3$ was not significant, and continuing to increase the ratio would further weaken the performance of the gates. In Figure 3, we calculated the change in azimuth as a function of the ratio. The results showed that as the ratio increased, the change in azimuth angle decreased continuously, where $|\beta_{H,T-}| = |\beta_2 - \beta_1|$. All things considered, $\eta = 2$ was an appropriate choice to construct quantum gates.

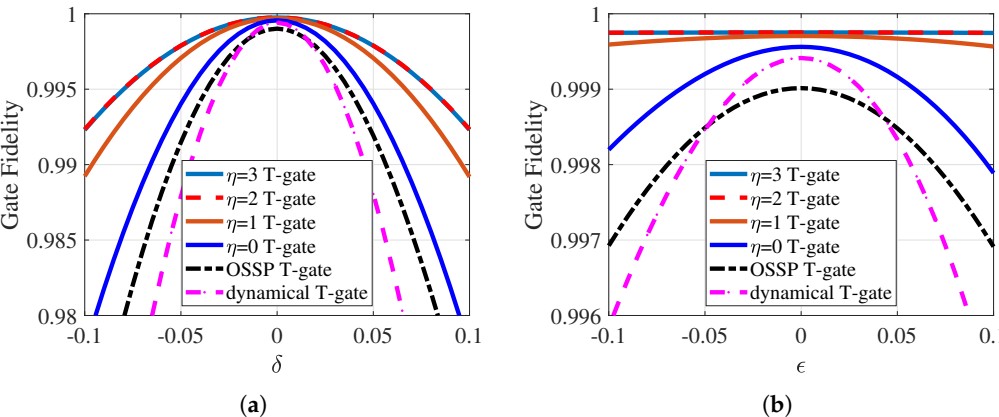

**Figure 4.** *Cont.*

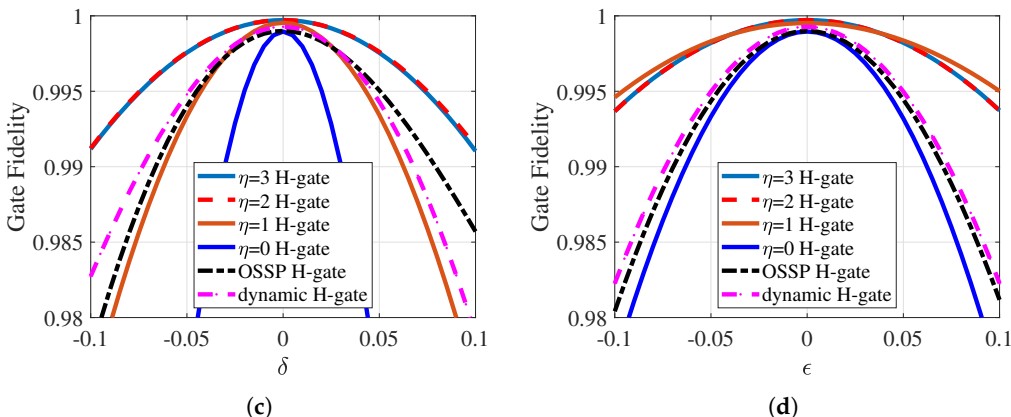

**Figure 4.** Test of the robustness of the $\eta = 0, 1, 2, 3$ T-gate and H-gate against control errors and comparison with the OSSP scheme and dynamical gates. The results show that we can improve the fidelity of the gate by choosing a specific ratio. (**a–c**) $\delta$ and (**b–d**) $\epsilon$ errors.

## 4. Implementation on Superconducting Circuits

In this section, we present the implementation of our ratio-optimized conventional geometric gates on superconducting quantum circuits. We first demonstrate the implementation of single-qubit gates in superconducting circuits and numerically calculate the fidelity of quantum gates. Then, we set out to realize and numerically verify the two-qubit gate on two capacitively coupled transmon qubits [37].

### 4.1. One-Qubit Gate

We started by constructing one-qubit gates in superconducting circuits. Because the Josephson junction is a nonlinear inductor, the energy spectrum of the transmon qubit can be equivalent to a two-level system, i.e., $H = \omega_1 \sigma_z / 2$, where $\omega_1$ is the transmon's resonance frequency. The two lowest levels were used as our logical qubit. Then, we added a microwave field to drive a single transmon, as in Figure 1b. The Hamiltonian read

$$H(t) = \sum_{k=1}^{\infty} \left[ k\omega_1 - \frac{1}{2}k(k-1)\alpha' \, |k\rangle \langle k| \right] + \left[ \frac{1}{2}\Omega(t)ae^{i\left[ \int_0^t \omega(t') \, dt' - \mu(t) \right]} + H.c \right] \tag{14}$$

where $\omega_1$, $\alpha'$, $\omega$, and $\mu$ are the qubit frequency, transmon's anharmonicity, microwave frequency, and phase, respectively. Here, the annihilation operator was always associated with the emission of photons, because we ignored the process of absorbing photons using the rotation wave approximation. The second excited state $|2\rangle$ was our main source of leakage, because of the anharmonicity of the transmon. Therefore, we truncated the annihilation operator to $a = |0\rangle \langle 1| + \sqrt{2} |1\rangle \langle 2|$. After rotating the Hamiltonian to the new framework with $U = U_2 U_1$, where $U_1 = \exp[-i\omega_1 at]$, and $U_2 = \exp[i \int_0^t \Delta(t') \, dt' (|1\rangle \langle 1| - |0\rangle \langle 0| + 3 |2\rangle \langle 2|)/2]$, the new Hamiltonian was obtained as

$$H(t) = \frac{1}{2}\{\Delta(t)(|1\rangle \langle 1| - |0\rangle \langle 0| + 3 |2\rangle \langle 2|)\} - \alpha' |2\rangle \langle 2|$$
$$+ \frac{1}{2}\{\Omega(t)e^{-i\mu(t)}(|0\rangle \langle 1| + \sqrt{2} |1\rangle \langle 2|) + H.c\} \tag{15}$$

where $\Delta(t) = \omega_1 - \omega(t)$. As in Figure 1b, the Rabi frequency could be adjusted by the xy control line and arbitrary waveform generator. The Z control line modified the detuning by changing the magnetic flux in the superconducting quantum interference device (SQUID).

We used the derivative removal via adiabatic gate (DRAG) technology to suppress the qubit leakage error caused by state $|2\rangle$ [38]. The corrected rabbi frequency was

$$\Omega_D(t) = \Omega(t) - \{i\dot{\Omega}(t) + [\dot{\mu}(t) + \Delta(t)]\Omega(t)\}/2\alpha' \tag{16}$$

The detailed derivation of the DRAG correction waveform is shown in Appendix D. According to a current state-of-art experiment [39], we set the decoherence rate and anharmonicity of the transmon as $\kappa = \kappa_- = \kappa_z = 4 \times 2\pi$kHz and $\alpha' = 2\pi \times 220$ MHz, respectively. Generally speaking, a strong Rabi frequency $\Omega(t)$ corresponds to a fast rotation speed. However, the coupling term $|1\rangle\langle 2|$ in the Hamiltonian did not allow us to arbitrarily increase $\Omega(t)$. Therefore, it was necessary for us to further discuss the choice of Rabbi frequency $\Omega(t)$. In the top subfigure of Figure 5, we plotted the fidelities of the T-gate ans H-gate as a function of the amplitude $\Omega_M$. As expected, the relationship between quantum gates fidelities and pulse strength was fluctuating. We calculated the peak of the curve, which corresponded to the highest fidelity and optimized Rabbi frequency. When $\Omega_M$ was taken as $\Omega_M = 31 \times 2\pi$ MHz , the fidelity of the T-gate reached the highest $F_T^G \cong 99.98\%$, and the highest fidelity of the H-gate was $F_H^G \cong 99.95\%$ when $\Omega_M = 12 \times 2\pi$ MHz. Let us consider a concrete example. Suppose the initial input state $|\psi\rangle = (|0\rangle + |1\rangle)/\sqrt{2}$, and $|\psi\rangle = |0\rangle$ for the T-gate and H-gate, the ideal final states were $(|0\rangle + e^{i\frac{\pi}{4}}|1\rangle)/\sqrt{2}$ and $(|0\rangle + |1\rangle)/\sqrt{2}$, respectively. We evaluated the quantum gates by the state population and fidelities defined by $F_s = Tr[\rho_i\rho_e]$, where $\rho_{i,e}$ is the density operator of the final ideal state and the actual final density operator. The state population and fidelity dynamics are shown in the bottom of Figure 5. Both state fidelities $F_s$ of the T-gate and H-gate reached 99.97%. The green dotted line represents the population probability at state $|2\rangle$. We found that it was a flat straight line about zero, which meant that information leakage was well suppressed.

Until now, we have successfully realized high-fidelity superconducting one-qubit gates. However, for the convenience of experiments, we generally set the detuning as a constant value. In Appendix E, we also calculate this case.

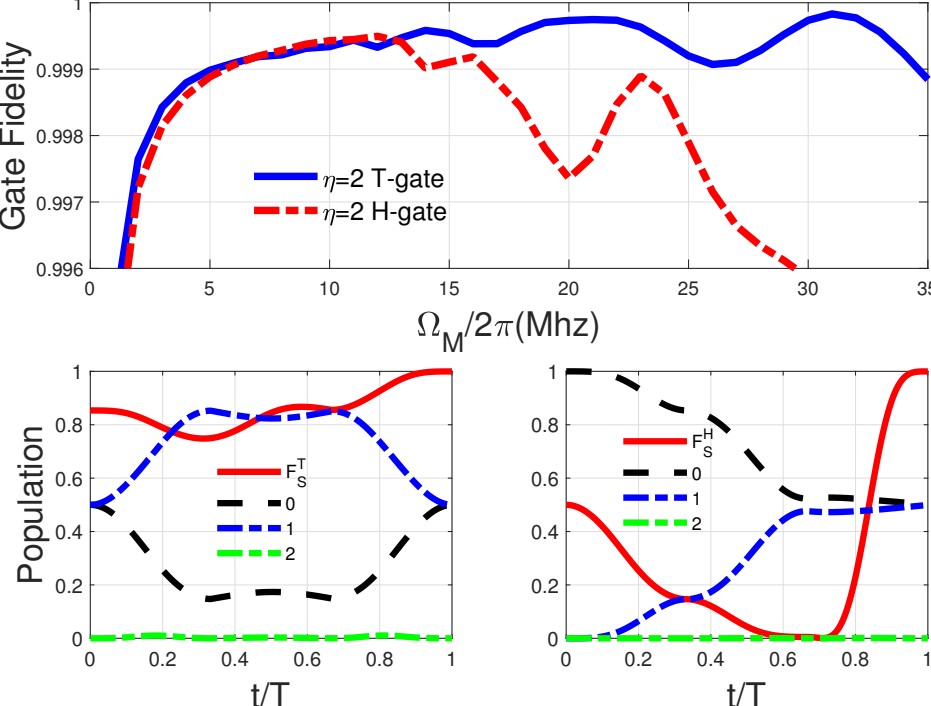

**Figure 5.** (**Top**) Gate fidelity as a function of the pulse peak $\Omega_M \in (0, 35]$ for the $\eta = 2$ T- and H-gate, the optimal value of $\Omega_M$ can be selected according to the highest gate fidelity. The state population and the state-fidelity dynamics for (**bottom and left**) $\eta = 2$ T-gate with input state $(|0\rangle + |1\rangle)/\sqrt{2}$ and (**bottom and right**) $\eta = 2$ H-gate with input state $|0\rangle$.

### 4.2. Two-Qubit Gate

We next discuss the construction of the ratio-optimized two-qubit gate on two directly capacitively coupled transmon qubits, labeled as $T_1$ and $T_2$, as in Figure 6. The Hamiltonian of the coupled system was written as

$$H_0(t) = \sum_{i=1,2} \sum_{j=1}^{+\infty} \left[ j\omega_i - \frac{j(j-1)}{2} \alpha_i' \right] |j\rangle_i \langle j| + g(a_1^\dagger - a_1)(a_2^\dagger - a_2), \tag{17}$$

where $\omega_j$ and $\alpha_j$ are the transmon's frequency and anharmonicity, respectively. In fact, the above formula is the general Jaynes–Cummings model. $g$ is the coupling strength, which is written as $g = C_g \sqrt{\omega_1 \omega_2} / \sqrt{(C_1 + C_g)(C_2 + C_g)}$. Once the quantum chip is prepared, the coupling strength and qubit frequency are fixed and not adjustable. Therefore, we needed to add external degrees of freedom to achieve a tunable coupling and efficient quantum control. Specifically, we added an external magnetic field $\epsilon(t)$ to modulate the SQUID frequency [40]. The corresponding Hamiltonian was

$$H(t) = H_0(t) + f(\epsilon)(|1\rangle_2 \langle 1| + |2\rangle_2 \langle 2|), \tag{18}$$

where $f(\epsilon)$ is the frequency response of the external field. Here, $f(\epsilon)$ was set to $f(\epsilon) = \dot{F}(t)$, where $F(t) = -\beta \cos[\nu t + \varphi(t)]$ with $\beta, \nu$, and $\varphi$ the amplitude, frequency, and phase, respectively.

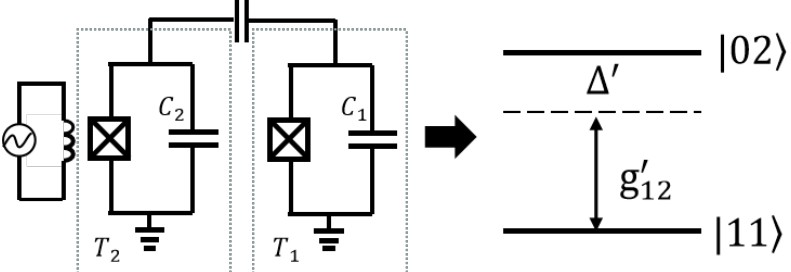

**Figure 6.** Schematic diagram of two directly capacitively coupled transmon qubits, labeled as $T_1$ and $T_2$, where $T_2$ is biased by an ac magnetic flux which periodically modulates its qubit frequency.

The modified qubit frequency was $\omega_2(t) = \omega_2 + \dot{F}(t)$. Then, we used

$$U_1 = \exp\left[ -i \sum_{i=1,2} \sum_{j=1}^{+\infty} \left[ j\omega_i - \frac{j(j-1)}{2} \alpha_i' \right] |j\rangle_i \langle j| \right]$$
$$U_2 = \exp[-iF(t)(|1\rangle_2 \langle 1| + |2\rangle_2 \langle 2|)] \tag{19}$$

to rotate the Hamiltonian picture. The new Hamiltonian was obtained as

$$H(t) = g\left[ |10\rangle \langle 01| e^{i\Delta_1 t} + \sqrt{2} |20\rangle \langle 11| e^{i(\Delta_1 - \alpha_1')t} + \sqrt{2} |11\rangle \langle 02| e^{i(\Delta_1 + \alpha_2')t} \right] e^{i\beta \cos(\nu t + \varphi(t))} + \text{H.c}, \tag{20}$$

where $\Delta_1 = \omega_1 - \omega_2$. We next handled the exponent part with the Jacobi–Anger identity,

$$e^{i\beta \cos\theta} = \sum_{n=-\infty}^{\infty} i^n J_n(\beta) e^{in\theta}, \tag{21}$$

in which $J_{-m}(\beta) = (-1)^m J_m(\beta)$, and $J_m(\beta)$ are Bessel functions of the first kind. Meanwhile, setting $\nu = \Delta_1 + \alpha_2' + \Delta'$ to obtain an off-resonance coupling with detuning $\Delta'$ between qubit $|11\rangle$ and $|02\rangle$ and neglecting the high-order oscillating terms by rotating-wave approximation yielded the effective Hamiltonian as

$$H = -\frac{1}{2}\Delta'(|11\rangle \langle 11| - |02\rangle \langle 02|) + \frac{1}{2}g'\left[ e^{-i\mu(t)} |11\rangle \langle 02| + \text{H.c} \right], \tag{22}$$

where $g' = 2\sqrt{2}gJ_1(\beta)$ is the effective coupling strength that can be modified by $\beta$, and the effective phase is $\mu(t) = \varphi(t) - \frac{\pi}{2}$. Obviously, the Hamiltonian in Equation (22) had the same form as Equation (11). Hence, the two excited states $|11\rangle$ would accumulate a total phase similarly to the one-qubit case, i.e., $|11\rangle \rightarrow \exp[-if(T)]|11\rangle$. The two-qubit computation space was $\mathcal{H} = span\{|11\rangle, |01\rangle, |10\rangle, |11\rangle\}$ and we could construct a controlled phase gate (CP) from the Hamiltonian in Equation (22) as

$$U(\tau) = |0\rangle\langle 0| \otimes I_{dim=2} + |1\rangle\langle 1| \otimes (|0\rangle\langle 0| + e^{-if(T)}|1\rangle\langle 1|). \tag{23}$$

Then, we tested the robustness of the CP gate by setting $f(T) = \pi/8$ and $\eta = 2$ as a typical example. The master equation could be written as

$$\dot{\rho} = -i[H(t), \rho] + \frac{1}{2}\sum_{j=1}^{2}\left\{\kappa_z^j \mathcal{L}(\mathcal{A}_z^j) + \kappa_-^j \mathcal{L}(\mathcal{A}_{-z}^j)\right\}, \tag{24}$$

where $\mathcal{A}_-^j = |0\rangle_j\langle 1| + \sqrt{2}|1\rangle_j\langle 2|$, and $\mathcal{A}_z^j = |1\rangle_j\langle 1| + 2|2\rangle_j\langle 2|$. For a general input state $|\psi_2\rangle = (\cos\theta_1|0\rangle_1 + \sin\theta_1|1\rangle_1) \otimes (\cos\theta_2|0\rangle_2 + \sin\theta_2|1\rangle_2)$, the ideal final state was $|\psi_2^f\rangle = $ CP$|\psi_2\rangle$. The two-qubit gate was defined as $F_2^G = \frac{1}{4\pi^2}\int_0^{2\pi}\int_0^{2\pi}\langle\psi_2^f|\rho|\psi_2^f\rangle\,d\theta_1 d\theta_2$ [34], and the integration was numerically performed for the 10001 initial states with $\theta_1$ and $\theta_2$ uniformly distributed in $[0, 2\pi]$ and setting the coupling strength $g = 2\pi \times 8$ MHz, $\alpha_2' = \alpha_1' = 2\pi \times 220$ MHz, and the decay and dephasing rates as $\kappa_-^{1,2} = \kappa_z^{1,2} = \kappa = 2\pi \times 4$ kHz [39]. In Figure 7 (Top), we plotted the gate fidelity as a function of $\Delta_1$ and $\beta$. The gate fidelity was $F_G \geq 99.8\%$ in the parameter area surrounded by white lines. For example, the gate fidelity reached 99.85% with $\beta = 1.6$, and $\Delta_1 = 315 \times 2\pi$ MHz. Furthermore, we calculated the state population and the state fidelity dynamics with the input state being $(|01\rangle + |11\rangle)/\sqrt{2}$ in Figure 7 (Bottom). Finally, the state fidelity of the CP gate reached 99.74%.

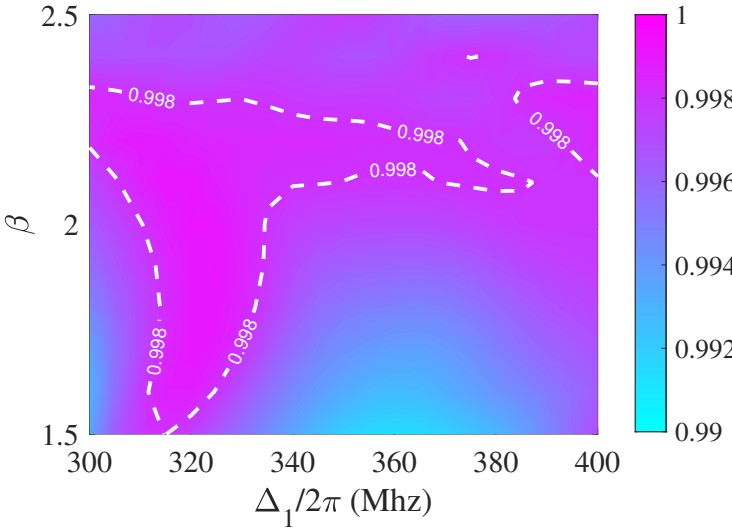

**Figure 7.** *Cont.*

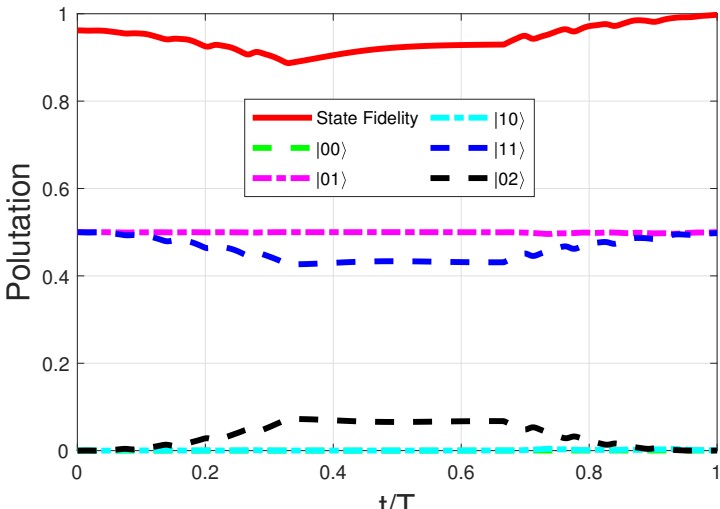

**Figure 7.** (**Top**) Gate fidelity of the nontrivial controlled phase gate as a function of parameters $\Delta_1$ and $\beta$ with $\eta = 2$. (**Bottom**) The state population and state-fidelity dynamics with initial state $(|01\rangle + |11\rangle)/\sqrt{2}$.

## 5. Conclusions

In this article, we proposed a general approach to realize an unconventional geometric quantum computation. Based on our method, the Hamiltonian could be reversely designed, thereby avoiding a complex parameter design in the forward direction. Moreover, the ratio between the phases could be adjusted arbitrarily in our method. Through a robustness test, we found that different ratios corresponded to different gate performances. Then, we optimized the ratio value to improve the robustness of quantum gates. Finally, we discussed the implementation of our ratio-optimized quantum gates in superconducting circuits. The results showed that the fidelities of the T-gate, Hadamard H-gate, and controlled phase gate in superconducting circuits were 99.98%, 99.95%, and 99.85%, respectively. The fidelity of these gates was high enough. Therefore, our scheme provides a promising way to realize large-scale fault-tolerant quantum computation in superconducting circuits.

**Author Contributions:** Conceptualization, Y.L.; data curation, Y.L.; investigation, Y.L.; writing—original draft, Y.L.; writing—review and editing, X.Z. All authors have read and agreed to the published version of the manuscript.

**Funding:** This work was supported by the National Natural Science Foundation of China (grant number. 61875060), the Key-Area Research and Development Program of Guangdong province (grant number. 2019B030330001).

**Institutional Review Board Statement:** Not applicable.

**Informed Consent Statement:** Not applicable.

**Data Availability Statement:** Not applicable.

**Conflicts of Interest:** The authors declare no conflict of interest.

## Appendix A. Derivation of the General Method

We give the specific steps of Equation (4)'s derivation in this appendix. We know that the Schrödinger equation is $i\frac{\partial}{\partial t}U = H \cdot U$. Suppose the Hilbert space is spanned by $\{|\tilde{\psi_1}\rangle, ..., |\tilde{\psi_L}\rangle\}$. Then, the evolution operator is written as $U = \sum_{i=1}^{L} |\psi_i(t)\rangle \langle \tilde{\psi_i}(0)|$ with

$|\psi_i(t)\rangle$ being the solution of the Schrödinger equation. Substituting the evolution operator into the Schrödinger equation gives

$$H = i\sum_{i=1}^{L} |\dot{\psi}_i\rangle \langle \psi_i| ; \tag{A1}$$

we next substitute Equations (1) and (3) into Equation (A1) and we have

$$
\begin{aligned}
H &= i\sum_{i=1}^{L}\Big[ i(1+\eta)\langle\tilde{\psi}_i| id_t |\tilde{\psi}_i\rangle |\tilde{\psi}_i\rangle\langle\tilde{\psi}_i| + |\dot{\tilde{\psi}}_i\rangle\langle\tilde{\psi}_i|\Big]\\
&= i\sum_{i=1}^{L}\Big[ -(1+\eta)\langle\tilde{\psi}_i|\dot{\tilde{\psi}}_i\rangle |\tilde{\psi}_i\rangle\langle\tilde{\psi}_i| + \sum_{l=1}^{L}\langle\tilde{\psi}_l|\dot{\tilde{\psi}}_i\rangle |\tilde{\psi}_l\rangle\langle\tilde{\psi}_i|\Big]\\
&= i\sum_{i=1}^{L}\Big[ -(1+\eta)\langle\tilde{\psi}_i|\dot{\tilde{\psi}}_i\rangle |\tilde{\psi}_i\rangle\langle\tilde{\psi}_i| + \sum_{k=i}^{L}\langle\tilde{\psi}_k|\dot{\tilde{\psi}}_i\rangle |\tilde{\psi}_k\rangle\langle\tilde{\psi}_i| + \sum_{i\neq l}^{L}\langle\tilde{\psi}_l|\dot{\psi}_i\rangle |\tilde{\psi}_l\rangle\langle\tilde{\psi}_i|\Big]\\
&= i\sum_{i\neq l}^{L}\langle\tilde{\psi}_l|\dot{\tilde{\psi}}_i\rangle |\tilde{\psi}_l\rangle\langle\tilde{\psi}_i| - i\eta\sum_{i=1}^{L}\langle\tilde{\psi}_i|\dot{\tilde{\psi}}_i\rangle |\tilde{\psi}_i\rangle\langle\tilde{\psi}_i|
\end{aligned}
\tag{A2}
$$

In the derivation process, we did not make any constraints on the auxiliary state $|\tilde{\psi}_i\rangle$ and the space dimension. Therefore, the formula Equation (A2) gives the general Hamiltonian for constructing control gates with any number of qubits. At the same time, this Hamiltonian is not only applicable to quantum computing and quantum information but also to general condensed matter problems. For example, the auxiliary state can be taken as the wave function of the phonon.

**Appendix B. OSSP Gate**

The evolution path of OSSP can be designed as follows. In $[0, T_1]$, the state evolves from the point $(\alpha_0, \beta_0)$ to the north pole along the geodesic $\beta(t) = \beta_0$. For this path, the Hamiltonian is written as $H(t) = \frac{1}{2}\Omega(t)[e^{-i(\beta-\pi/2)}|0\rangle\langle 1| + \text{H.c}]$, where the Rabi frequency $\Omega(t) = \dot{\alpha}(t)$. Accordingly, the pulse is $\int_0^{T_1}\Omega(t)\,dt = \alpha_0$. Then, the state evolves form the north pole to the south pole along the geodesic $\beta(t) = \beta_1$ in $(T_1, T_2]$, the Hamiltonian is expressed as $H(t) = \frac{1}{2}\Omega(t)[e^{-i(\beta+\pi/2)}|0\rangle\langle 1| + \text{H.c}]$. The pulse is $\int_{T_1}^{T_2}\Omega(t)\,dt = \pi$. Finally, returning back to the point $(\alpha_0, \beta_0)$ along geodesic $\beta(t) = \beta_0$ in $(T_2, T]$, the Hamiltonian is written as $H(t) = \frac{1}{2}\Omega(t)[e^{-i(\beta-\pi/2)}|0\rangle\langle 1| + \text{H.c}]$, and the pulse area is equal to $\pi - \alpha_0$. The pure geometric phase of the OSSP can be calculated as $\gamma_g = \beta_0 - \beta_1$. Accordingly, the evolution operator is

$$U(T) = \cos\gamma_g - i\sin\gamma_g \begin{pmatrix} \cos\alpha_0 & \sin\alpha_0 e^{-i\beta_0} \\ \sin\alpha_0 e^{i\beta_0} & -\cos\alpha_0 \end{pmatrix}. \tag{A3}$$

The dynamical phase is equal to zero when the evolution path is geodesic. When $\alpha_0 = 0$, $\gamma_g = \pi/8$, the T-gate can be realized, and if $(\alpha_0, \beta_0) = (\pi/4, 0)$, and $\gamma_g = \pi/2$, the H-gate can be realized.

**Appendix C. Dynamical Gate**

The Hamiltonian of the two-level quantum system resonantly driven by an external field can be written as

$$H(t) = \frac{1}{2}\Omega(t)[\cos\mu(t)\sigma_x + \sin\mu(t)\sigma_y]. \tag{A4}$$

Setting $\mu(t) = \mu_d$ to make $[H(t_1), H(t_2)] = 0$ is fulfilled. Therefore, the evolution operator is obtained as

$$\begin{aligned} U(\theta_d, \mu_d) \quad &= \exp[-i \int_0^T H(t)\, dt] \\ &= \begin{pmatrix} \cos(\theta_d/2) & -i\sin(\theta_d/2)e^{-i\mu_d} \\ -i\sin(\theta_d/2)e^{i\mu_d} & \cos(\theta_d/2) \end{pmatrix}, \end{aligned}$$

where $\theta_d = \int_0^T \Omega(t)\, dt$, and the constant $\mu_d$ can ensure the geometric phase is zero. The arbitrary dynamical X, Y, and Z rotating gate can be realized by $R_x(\theta_x) = U_d(\theta_x, 0)$, $R_y(\theta_y) = U_d(\theta_y, \pi/2)$, and $R_z(\theta_z) = U_d(\pi/2, \pi)U_d(\theta_z, -\pi/2)U_d(\pi/2, 0)$, respectively. In this way, the T gate and H gate can be realized as $R_z(\pi/4)$ and $U_d(\pi, \pi)U_d(\pi/2, \pi/2)$, respectively.

**Appendix D. DRAG Correction**

We here give the details of the calculation of the DRAG correction waveform in Equation (16). In fact, because the coupling of higher-excited states is not negligible, we cannot get the ideal two-level model. Here, we consider the influence of the second excited state, which is the main leakage source due to the anharmonicity. Therefore, we apply the DRAG technology to suppress the information leakage. To this end the Hamiltonian describing the three-level anharmonic oscillator is written as

$$H(t) = \frac{1}{2}\mathbf{B}(t) \cdot \mathbf{S} + \delta \,|2\rangle \langle 2| \tag{A5}$$

where $\delta$ is an anharmonic parameter. The vector operator $\mathbf{S}$ is given by

$$\begin{aligned} S_x &= \sum_{0,1} \sqrt{n+1}(|n+1\rangle \langle n| + h.c) \\ S_y &= \sum_{0,1} \sqrt{n+1}(i\,|n+1\rangle \langle n| + h.c) \\ S_z &= \sum_{0,1,2} (1-2n)\,|n\rangle \langle n| \end{aligned} \tag{A6}$$

The vector $\mathbf{B} = \mathbf{B}_0 + \mathbf{B}_d$ is the total controlled microwave field, where $\mathbf{B}_0 = (B_x, B_y, B_z)$ and $\mathbf{B}_d = (B_{d,x}, B_{d,y}, B_{d,z})$ are the initial microwave field and the DRAG correction term. The correction term is written as

$$\begin{aligned} B_{d,x} &= \frac{1}{2\delta}\big[\dot{B}_y(t) - B_z(t)B_x(t)\big] \\ B_{d,y} &= -\frac{1}{2\delta}\big[\dot{B}_x(t) + B_z(t)B_y(t)\big] \\ B_{d,z} &= 0 \end{aligned} \tag{A7}$$

According to the Hamiltonian Equation (15), we have $\mathbf{B}_0 = (\Omega(t)\cos\mu(t), \Omega(t)\sin\mu(t), -\Delta)$ and $\delta = \alpha'$. We obtain that

$$\begin{aligned} B_{d,x} &= -\frac{1}{2\alpha'}\big[\dot{\Omega}\sin\mu + \Omega\dot{\mu}\cos\mu + \Delta\Omega\cos\mu\big] \\ B_{d,y} &= \frac{1}{2\alpha'}\big[\dot{\Omega}\cos\mu - \Omega\dot{\mu}\sin\mu - \Delta\Omega\sin\mu\big] \\ B_{d,z} &= 0 \end{aligned} \tag{A8}$$

Accordingly, the corrected Hamiltonian is

$$H(t) = \frac{1}{2}\{\Delta(t)(|1\rangle\langle 1| - |0\rangle\langle 0| + 3|2\rangle\langle 2|)\} - \alpha'|2\rangle\langle 2|$$
$$+ \frac{1}{2}\{\Omega_D(t)e^{-i\mu(t)}(|0\rangle\langle 1| + \sqrt{2}|1\rangle\langle 2|) + \text{H.c}\} \tag{A9}$$

and the corrected Rabi frequency is written as

$$\Omega_D(t) = \Omega(t) - \{i\dot{\Omega}(t) + [\dot{\mu}(t) + \Delta(t)]\Omega(t)\}/2\alpha' \tag{A10}$$

## Appendix E. Constant Detuning

For the convenience of experiments, we often set the detuning amount as a constant. In section three, we obtained $\Delta(t) = 0$ and $\dot{\mu}(t)$, when the quantum state evolved along the longitude line. For the latitude line, the detuning and the Rabi frequency are written as $\Delta(t) = -2[1 + \frac{\sqrt{2}}{2}(1 + \eta)]\sin^2\frac{\pi}{8}\dot{\beta}(t)$, and $\Omega(t) = \frac{\sqrt{2}}{2}[\frac{\sqrt{2}}{2}(1 + \eta) - \eta]\dot{\beta}(t)$, respectively. If the Rabi frequency is a square pulse, then the $\dot{\beta}$ and $\Delta$ are both constants. However, we obtain that the correct pulse under DRAG is $\Omega_D(t) = \Omega(t) - \{i\dot{\Omega}(t) + [\dot{\mu}(t) + \Delta(t)]\Omega(t)\}/2\alpha'$. Meanwhile, when $\Omega(t)$ is a square pulse, i.e., $\dot{\Omega}(t) = 0$, the correct pulse becomes $\Omega_D(t) = \Omega$. In other words, the DRAG technology is invalid. Therefore, the Rabi frequency was only set as square pulse $\Omega_M$ along the latitude line. After calculation, when $\Omega_M = 32 \times 2\pi MHz$ and $\eta = 2$, we obtained the fidelity of the T-gate as 99.96%, and when $\Omega_M = 13 \times 2\pi MHz$ and $\eta = 1.9$, the fidelity of the H-gate reached 99.95%.

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
