# Peer review of "Optimized Unconventional Geometric Gates in Superconducting Circuits"

_applsci, doi:10.3390/app13064041_

Round 1

Reviewer 1 Report

The authors develop an alternative approach to designing geometric gates and apply the technique to superconducting qubits. The problem is well motivated and the manuscript is clearly written. The results on CP gates are especially interesting. Their predicted fidelities are close to Google's values (of course this also depends on their decoherence model). 

I can recommend publication after the authors do a direct comparison with Google's CZ gate protocol (also based on third level repulsion). See "Superconducting quantum circuits at the surface code threshold for fault tolerance" (doi:10.1038/nature13171). Namely, compare the geometric and Google approaches, both optimized with in the presence of the same decoherence model. This would be very valuable to the superconducting qubit literature.

Author Response

Please see the attachment for the specific reply.

Reviewer 2 Report

Quantum computing becomes more and more important as a new research study.  Reliable and effective computational techniques are important.
  This paper is valuable for the readers: Sec. 1 is a good review including up to date information. Sec.2 includes clear description of the method, etc.

There are several unclear places for me: 2.2. at the first line; f(0). f_i(0) ? After Eq.10; Rabi or Rabbi ?  The 3rd line in Sec.5; avoiding avoiding (Why doubled ? Want to stress?)

Author Response

(The authors gave the same response as above.)

Reviewer 3 Report

Geometric phases are known for their invariance with respect to gauge

transformation. Quantum computation using geometric phases is thereby

considered to inherit such robustness.  In this paper the authors propose an

approach to realize  generalized unconventional geometric computation,

which can help to construct  higher-fidelity gates. I think the paper is interesting and valuable. The manuscript is also well prepared.  I would recommend the publication of this manuscript.

Author Response

(The authors gave the same response as above.)
